# Genetic and Clinical Spectrum of Limb–Girdle Muscular Dystrophies in Western Sicily

**DOI:** 10.3390/genes16080987

**Published:** 2025-08-21

**Authors:** Nicasio Rini, Antonino Lupica, Paolo Alonge, Grazia Crescimanno, Antonia Pignolo, Christian Messina, Sandro Santa Paola, Marika Giuliano, Eugenia Borgione, Mariangela Lo Giudice, Carmela Scuderi, Vincenzo Di Stefano, Filippo Brighina

**Affiliations:** 1Department of Biomedicine, Neuroscience, and Advanced Diagnostic (BIND), University of Palermo, 90129 Palermo, Italy; nicasio.rini@unipa.it (N.R.); antlupica@gmail.com (A.L.); alongep95@gmail.com (P.A.); grazia.crescimanno@irib.cnr.it (G.C.); antonia.pignolo@unipa.it (A.P.); chry.messina@gmail.com (C.M.); filippobrighina@gmail.com (F.B.); 2Oasi Research Institute-IRCCS, Via Conte Ruggero 73, 94018 Troina, Italy; ssantapaola@oasi.en.it (S.S.P.); mgiuliano@oasi.en.it (M.G.); eborgione@oasi.en.it (E.B.); mlogiudice@oasi.en.it (M.L.G.); cscuderi@oasi.en.it (C.S.)

**Keywords:** limb–girdle muscular dystrophies, calpainopathy, *CAPN3*, *DYSF*, *LAMA2*, *ANO5*, *FKTN*, *TTN*

## Abstract

**Background and Objectives:** Limb–girdle muscular dystrophies (LGMDs) are a group of muscular dystrophies characterized by predominantly proximal-muscle weakness, with a highly heterogeneous genetic etiology. Despite recent efforts, the epidemiology of LGMDs is still under-evaluated. However, a better understanding of the distribution and genetic characteristics of LGMDs is required to optimize the diagnostic process and to address future research. Therefore, the aim of the present study is to investigate and identify new pathogenic variants, to better characterize LGMDs in Sicily. **Methods:** We enrolled patients with genetic and clinical diagnosis of LGMD referred to our clinic between the years 2019 and 2025. A targeted next-generation-sequencing (NGS) panel was performed, based on the reported disease frequency. A retrospective analysis of the clinical, laboratory, electrophysiological, and histological features was performed. **Results:** A total of 28 LGMDs patients aged 56.6 years (47.2–60.5 IQR) were identified (16 males, 57%). A molecular diagnosis was achieved in 24 (85.7%) of patients, most commonly carrying mutations in *CAPN3* (14 patients, 50%), followed by *DYSF*, *LAMA2*, *ANO5*, *FKTN* and *TTN* genes. Pathogenic variants in *CAPN3* and *LAMA2* were associated with earlier onset and longer disease duration, whereas *ANO5* presented later with a milder course. Cardiac involvement was observed more frequently in patients with *LAMA2* and *FKTN mutations*. Association between heterozygous mutations in the CAPN3 and DYSF, as well as between CAPN3 and DMD variants were reported. **Discussion:** The findings of this study provide valuable insights into the epidemiology of LGMDs in the Western Sicily, offering important contributions to genotype–phenotype correlations. Our analysis highlights the role of genetic diagnosis in achieving accurate classification of the disease and optimizing clinical management.

## 1. Introduction

Limb–girdle muscular dystrophies (LGMDs) represent a genetically and clinically heterogeneous group of muscular dystrophies involving the pelvic and shoulder girdle muscles that share progressive muscle weakness and dystrophic features on muscle biopsy [1,2,3,4,5]. Originally described by Walton and Nattrass in 1954 to distinguish them from X-linked dystrophinopathies, LGMDs now encompass more than 29 distinct genetic subtypes, each associated with pathogenic variants in genes involved in muscle structure, maintenance, or repair [1,2,3,4,5,6,7]. Among the genes most frequently involved in LGMDs, *CAPN3* encodes calpain-3, a muscle-specific protease that plays a crucial role in sarcomere remodeling and muscle fiber stability [8]. *DYSF* encodes dysferlin, a membrane-associated protein essential for repairing muscle cell damage [9]. *LAMA2* produces the laminin-α2 chain, a structural component of the basal lamina that supports muscle integrity [10]. *ANO5* is believed to regulate calcium-dependent membrane repair, while *FKTN* is involved in the glycosylation of alpha-dystroglycan, a process fundamental to sarcolemmal stability [11,12]. Finally, *TTN* encodes titin, a giant protein responsible for the passive elasticity and structural organization of muscle sarcomeres [13]. Mutations in these genes can compromise various cellular processes that are crucial for muscle homeostasis, leading to progressive muscle degeneration. The LGMD nomenclature was revised in 2018 [2], introducing a classification system that emphasizes inheritance patterns, categorizing LGMDs into autosomal dominant (LGMD-D) and autosomal recessive (LGMD-R) forms. The recessive types (previously designated as LGMD2) account for approximately 90% of cases and typically present earlier with more rapid progression than dominant forms, representing fewer than 10% of LGMD cases [7,14]. The global prevalence of LGMDs is estimated to range from 0.8 to 6.9 per 100,000 individuals, with *CAPN3*-related calpainopathy (LGMD2A/R1) emerging as the most common subtype, followed by *DYSF*-related LGMD (LGMDR2) [7,14,15]. The Italian LGMD registry reported the *CAPN3* gene as the leading cause, with *DYSF* and sarcoglycanopathies also frequently represented, reflecting a distribution pattern comparable to other European countries [16]. In contrast, *FKRP*-related LGMD2I/R9 and *ANO5*-related LGMD (LGMDR12) are both more frequently observed in Northern Europe and among certain founder populations, consistent with well-documented geographic clusters [17,18,19].

The extensive genetic heterogeneity of LGMDs, combined with overlapping phenotypes with other myopathies, continues to pose considerable diagnostic challenges. Although next-generation sequencing (NGS) technologies have greatly improved the diagnostic approach by enabling simultaneous analysis of multiple genes associated with LGMD, a substantial proportion of patients remain genetically undiagnosed, and many harbor variants of uncertain significance (VUS) [7,14]. Consequently, a comprehensive diagnostic work-up often includes a combination of clinical assessment, serum creatine kinase (CK) measurement, electromyography (EMG), muscle MRI to detect characteristic patterns of fatty infiltration, and in selected cases, muscle biopsy to demonstrate dystrophic changes [20,21]. Despite these advances, no disease-modifying therapies are currently approved for LGMDs, and management remains primarily supportive, relying on a multidisciplinary approach to preserve ambulation and respiratory function and to delay cardiac or orthopedic complications. Nonetheless, improvements in standards of care and anticipatory strategies have significantly altered the natural history of muscular dystrophies, enhancing both survival and quality of life. Additionally, promising therapeutic avenues such as systemic gene replacement and RNA-modulating strategies are currently under investigation for specific LGMD subtypes [1,2,3]. In Southern Europe, data on the distribution and molecular spectrum of LGMDs remain limited, particularly in genetically distinct regions such as Sicily. Of interest, the worldwide distribution of LGMDs is influenced by several factors, including orogeographical characteristics, as reported in the Reunion Island [22,23]. Indeed, historical patterns of migration and consanguinity in Sicily may have contributed to unique mutational landscapes and higher local frequencies of specific LGMD subtypes. Characterizing the genetic and clinical spectrum of LGMDs in such underrepresented populations is crucial to facilitate a timely molecular diagnosis and appropriate genetic counseling for affected families. Still, despite recent efforts to create large descriptions of cohorts and specific disease registries, prevalence estimation studies suggest that LGMDs are still underdiagnosed [15]. Therefore, the present study aimed to investigate the genetic and clinical spectrum of LGMDs in Western Sicily. By analyzing a cohort of patients evaluated at neuromuscular centers across the region, we sought to delineate the relative prevalence of LGMD subtypes, characterize their phenotypic presentations, and assess the impact of genetic findings on clinical management. Through this effort, we hope to contribute to a more comprehensive understanding of LGMDs and to support future initiatives directed at precision diagnosis and care for these debilitating disorders.

## 2. Materials and Methods

### 2.1. Study Design and Patients Population

A retrospective analysis was performed on 28 patients recruited at the Regional Center for Neuromuscular Rare Disease at the University of Palermo and the Oasi Research Institute (Troina) with a clinical diagnosis of limb–girdle muscular dystrophy (LGMD), established based on clinical features, electromyography, muscle biopsy findings and/or genetic testing. For each patient included, clinical, laboratory, electrophysiological, radiological, and histological data were systematically collected, as well as age at onset, gender, parental consanguinity, family history of neuromuscular disease, presenting symptoms, and disease course. All studies were conducted in accordance with ethical principles, and informed consent for genetic testing was obtained from all patients either in the outpatient clinic or during hospital admission for diagnostic purposes. Muscle strength was assessed by manual muscle testing and graded from 0 to 5 according to the Medical Research Council scale. Whenever available, data from muscle biopsies were collected, and findings were categorized as dystrophic (consistent with muscular dystrophies), non-dystrophic myopathic (e.g., non-specific changes), or normal. Electromyography (EMG) patterns were classified as normal, myopathic, neurogenic, or mixed (displaying both myopathic and neurogenic features or “neurogenic-like” patterns), in accordance with standard criteria [24]. Muscle MRI findings were categorized as either normal or dystrophic, based on the presence of fatty replacement of muscle tissue on imaging.

### 2.2. Molecular Analysis

Genomic DNA was extracted from peripheral blood leukocytes of patients and available family members using standard procedures. Molecular testing was performed by targeted next-generation sequencing (NGS) on the NextSeq 550 System platform (Illumina, San Diego, CA, USA), using the Illumina Exome Panel, which included comprehensive coverage of 36 genes associated with limb–girdle muscular dystrophies and related myopathies (*ANO5*, *BVES*, *CAPN3*, *CAV3*, *DAG1*, *DES*, *DMD*, *DNAJB6*, *DOK7*, *DYSF*, *FKRP*, *FKTN*, *GAA*, *GMPPB*, *LAMA2*, *LMNA*, *MYOT*, *PLEC*, *POMGNT1*, *POMGNT2*, *POMK*, *POMT1*, *POMT2*, *POPDC3*, *SEPN1*, *SGCA*, *SGCB*, *SGCD*, *SGCG*, *TCAP*, *TNPO3*, *TOR1AIP1*, *TRAPPC11*, *TRIM32*, *TTN*).

Sequencing data were aligned to the human reference genome (GRCh37/hg19) using the Burrows-Wheeler Aligner (BWA, version 0.7.17) and processed with the Genomic Analysis Toolkit (GATK) for variant calling. The average sequencing depth across target regions was 97%, with a minimum requirement of 20× coverage for reliable variant interpretation.

Raw variant data were annotated using the wANNOVAR web server (https://wannovar.wglab.org/ accessed on 14 July 2025), and filtered to exclude variants with a minor allele frequency (MAF) greater than 1% in population databases such as dbSNP, the 1000 Genomes Project (https://www.internationalgenome.org/1000-genomes-summary accessed on 14 July 2025), and Genome Aggregation Database (gnomAD, https://gnomad.broadinstitute.org/ accessed on 14 July 2025). Variants reported by ClinVar (https://www.ncbi.nlm.nih.gov/clinvar/ accessed on 14 July 2025) as benign or likely benign, as well as synonymous variants without predicted impact on splicing, were also excluded.

The variants (such as pathogenic, likely pathogenic, or variants of uncertain significance) were classified according to the guidelines of the American College of Medical Genetics and Genomics (ACMG). Sequence variations were compared with data available in the Human Gene Mutation Database (HGMD) and ClinVar.

The pathogenic potential of non-synonymous single nucleotide variants was further evaluated using multiple in silico prediction tools, including SIFT (http://sift.bii.a-star.edu.sg/ accessed on 14 July 2025), PolyPhen-2 (http://genetics.bwh.harvard.edu/pph2/ accessed on 14 July 2025), and Mutation Taster (http://www.mutationtaster.org/ accessed on 14 July 2025). Candidate variants were visually inspected with Integrative Genomics Viewer (IGV) to assess sequencing quality and alignment and were subsequently validated by Sanger sequencing to confirm the NGS findings and exclude technical artifacts. It should be noted that despite the high sensitivity and specificity of the methods employed, this approach does not detect large deletions or duplications, and certain genomic regions may not be adequately covered due to the presence of pseudogenes or high sequence homology. Additionally, the in silico predictions were interpreted as supportive evidence rather than definitive proof of pathogenicity.

### 2.3. Statistical Analysis

Descriptive statistics were used to summarize demographic, clinical, biochemical, and genetic data. Continuous variables were tested for normality using the Shapiro–Wilk test. Variables with a normal distribution were reported as mean ± standard deviation (SD), while those with a non-normal distribution were expressed as median and interquartile range (IQR). Categorical variables were presented as counts and percentages. Comparisons of continuous variables between groups were conducted using Student’s *t*-test for normally distributed variables and the Mann–Whitney U test for non-normally distributed data. For comparisons across multiple groups, one-way ANOVA or the Kruskal–Wallis test was used, depending on data distribution. Comparisons of frequencies across groups were performed using the chi-square test (χ^2^) or Fisher’s exact test, as appropriate. A *p*-value < 0.05 was considered statistically significant. All analyses were performed using SPSS Statistics version 30.0 (IBM Corp, Armonk, NY, USA).

## 3. Results

### 3.1. Patients’ Demographic and Clinical Features

A total of 28 patients with clinical and/or molecular diagnosis of limb–girdle muscular dystrophy (LGMD) were included, including 16 males (57%) and 12 females (43%), as shown in Table 1.

The mean age was 52.0 ± 16.3 years, with a median of 56.6 years (IQR: 47.2–60.5). The age at disease onset was 25.6 ± 17.7 years, with a median of 22.0 years (IQR: 15.0–34.0 years), while the disease duration showed a mean of 25.1 ± 18.1 years (median 21.0, IQR: 8.0–40.0). Serum CK levels were highly variable, with a median of 2000 U/L (IQR: 521.5–2649). Muscle biopsy revealed dystrophic or myopathic changes in 11 patients (39%), EMG was myopathic in 13 (46%), and muscle MRI showed fatty degeneration in 1 patient (4%). Figure 1 displays the most common clinical manifestations included shoulder weakness (75%), pelvic weakness (71%), scapular atrophy (64%), posterior leg weakness (68%), and anterior leg weakness (64%). Myopathic gait was observed in 15 patients (54%), joint or tendon contractures in 10 (36%), hand weakness in 8 (29%), and Gowers’ sign in 2 (7%). Less frequent findings included scapular winging (25%), calf hypertrophy (18%), lordosis (21%), and scoliosis (11%). Systemic involvement was documented in 5 patients (18%) with cardiac manifestations and 10 patients (36%) requiring respiratory support. Wheelchair dependency was reported in 10 patients (36%).

### 3.2. Genetic Findings and Genotype–Phenotype Correlations

Twenty-four patients achieved a molecular diagnosis of LGMDs. Pathogenic or likely pathogenic variants were identified in 19 patients, while 5 patients carried variants of uncertain significance (VUS). The most frequent genetic subtype was *CAPN3*-related LGMD (58%), followed by *DYSF* (12.5%), *LAMA2* (12.5%), *ANO5* (8.3%), *FKTN* (8.3%), and *TTN* (8.3%). Table 2 and Figure 2 describes the genotype–phenotype correlations in the cohort studied.

Patients with mutations in the *CAPN3* gene (n = 14) had a mean onset age of approximately 20 years, moderate CK elevation (~1908 U/L), frequent myopathic gait (57%), contractures (50%), and wheelchair dependency in 7 of 14 cases. Cardiac and respiratory involvement were rare. Patients with mutations in the *DYSF* gene (n = 5) showed a later onset (~28 years), the highest mean CK (~4000 U/L), occasional respiratory involvement, and no cardiac findings. Patients with mutations in the *LAMA2* gene (n = 3) displayed early onset (~19 years), prolonged disease duration (~44 years), lower CK (~383 U/L), cardiac involvement in two cases, and one requiring respiratory support. Patients with mutations in the *ANO5* gene (n = 2) had late onset (~48.5 years), short disease duration (~6.5 years), high CK (~2650 U/L), and no cardiac or respiratory involvement.

Patients with mutations in the *FKTN* gene (n = 2) presented in the third decade (~32.5 years) with CK~2168 U/L, preserved respiratory function, but both cases showed cardiac involvement consistent with dystroglycanopathy-related cardiomyopathy. Patients with mutations in the *TTN* gene (n = 2) exhibited variable onset, moderate CK (~750 U/L), Gowers’ sign, contractures, wheelchair dependence, and in one case respiratory involvement. Table 3 and Figure 3 report all genotypes encountered in the cohort of LGMDs patients.

Two patients presented a digenic combination of heterozygous mutations in LGMDs genes: a patient with a compound heterozygous mutation in the *CAPN3* gene associated with heterozygous mutation in the *DYSF* gene (Table 3, patient 23) and another patient with a compound heterozygous mutation in the *DYSF* associated with heterozygous mutation in the *CAPN3* gene (Table 3, patient 24). A further patient presented a combination of a heterozygous mutation in the *CAPN3* gene associated with heterozygous deletion from the 45 to the 48 exons in the *DMD* gene (Table 3, patient 9).

Patients with digenic combinations of LGMDs variants displayed more severe phenotypes. Patient 23 presented with a highly severe LGMD with onset at about 20 years of age, with loss of ambulation at 42 years and respiratory insufficiency with ventilatory dependence at 48 years. Her sister (Table 3, patient 6) and brother (Table 3, patient 7) presented milder symptoms and disability. Patient 24 presented a severe LGMD with onset at about 26 years of age, with elevated CPK (about 5000 U/L), myopathic changes at EMG and muscle biopsy. He presented rapidly progressive tetraparesis and respiratory insufficiency with tracheostomy and invasive ventilation with death at 56 years as a complication of respiratory insufficiency and sepsis. Patient 9 presented a proximal and progressive muscle weakness at four limbs with onset at 30 years with mild increased serum CK (about 500 U/L) and diffuse myopathic EMG changes.

### 3.3. Comparison Among Genotypes

A one-way ANOVA revealed a significant difference in age at the disease onset using as factor genetic subgroups. Games-Howell post hoc tests indicated that patients with *CAPN3* variants had a significantly earlier onset compared to those with *ANO5* variants (*p* < 0.001). Similarly, there was a significant difference in disease duration across genetic groups, with ANO5 showing a significantly shorter duration than *CAPN3* (*p* < 0.01) and *DYSF* (*p* < 0.036). In contrast, a Kruskal–Wallis test did not show a statistically significant difference in CK levels across the genetic subgroups (H(5) = 5.21, *p* = 0.391). When comparing genotype status, an independent samples t-test showed no significant difference in age at onset between patients with pathogenic variants and those with VUS (t(23) = −0.193, *p* = 0.848). A Mann–Whitney U test did not reveal a significant difference in CK levels between pathogenic variants (mean rank = 10.8) and VUS (mean rank = 7.0) (U = 21.0, *p* = 0.159). Finally, wheelchair dependency was significantly more frequent in patients with pathogenic variants (53% vs. 0%). The chi-square test confirmed this association (χ^2^(1) = 5.26, *p* = 0.022), although Fisher’s exact test showed borderline significance (*p* = 0.051), which warrants cautious interpretation due to small sample sizes.

## 4. Discussion

In this study, we provided a detailed clinico-genetic characterization of 28 patients with limb–girdle muscular dystrophy (LGMD) from Western Sicily, a population with a distinct genetic profile influenced by regional consanguinity and migration patterns. This represents one of the first studies in Southern Italy, complementing data from larger European cohorts and offering unique regional insights. 

Consistently with previous studies [1,15,16,25,26], our data confirm the predominance of *CAPN3*-related calpainopathy, accounting for half of our cases. These patients exhibited earlier onset, moderate hyperCKemia, frequent contractures and a substantial risk of wheelchair dependence. Our findings reinforce the established genotype–phenotype correlations for calpainopathy, characterized by early pelvic and posterior thigh involvement followed by progressive shoulder girdle weakness [27]. Interestingly, *DYSF* variants were the second most common genotype, in line with data from the Italian LGMD registry and other European cohorts [16,28]. Patients with *DYSF* mutations typically showed a later disease onset, the highest CK levels, and occasionally respiratory involvement, though none exhibited cardiac disease—again mirroring typical dysferlinopathy features [28,29,30]. A notable aspect of our cohort was the identification of three patients with *LAMA2* variants, all presenting with adult-onset of disease, low CK levels, and remarkably long disease duration, with cardiac involvement in two cases. These findings align with emerging literature describing milder, late-onset *LAMA2*-related phenotypes confined to skeletal and cardiac muscle, diverging from the classical congenital muscular dystrophy (MDC1A) presentation with white matter changes [31,32]. Our data support including *LAMA2* variants in the differential diagnosis of LGMD with cardiac involvement, even in adults. Similarly, the two patients with *FKTN* variants underscore the expanding phenotype associated with this gene. Both exhibited adult-onset limb–girdle weakness, elevated CK, and cardiac manifestations, without cognitive or structural brain involvement. This highlights that fukutin variants, beyond causing Fukuyama congenital muscular dystrophy, can underlie LGMD phenotypes with dystroglycanopathy-related cardiomyopathy [12,17,33]. These observations emphasize the importance of systematic cardiac surveillance in FKTN patients. Two patients harbored *TTN* variants, presenting with diverse onset ages, moderate CK elevation, joint contractures, and respiratory involvement. This heterogeneity is characteristic of titinopathies, which encompass a wide spectrum from congenital forms to adult-onset limb–girdle presentations [34,35]. A smaller subgroup of *ANO5*-related LGMD in our cohort showed the latest onset, high CK, short disease duration, and no cardiac or respiratory involvement. This aligns with previously described anoctaminopathies, often presenting as milder forms or even isolated hyperCKemia in Northern European and Italian cohorts [18,19,36].

From a clinical perspective, our study revealed significant differences in age at onset and disease duration across genetic subgroups. Patients with *ANO5* variants exhibited significantly later onset and shorter disease duration compared to *CAPN3* and *DYSF* patients, emphasizing the relatively indolent course and late presentation of *ANO5*-related myopathy. However, CK levels did not differ significantly between groups, reflecting the known overlap in biochemical profiles across LGMD subtypes. Importantly, our analysis highlighted a significantly longer disease duration and higher rate of wheelchair dependence in patients with pathogenic variants compared to those with VUS. This suggests that patients with genetically confirmed pathogenic variants tend to follow a more classic progressive disease course, whereas those with VUS might represent either milder forms or cases with undetected modifying factors.

This study also addresses the rare issue of overlapping syndromes involving different genes among LGMDs and between single LGMDs and other monogenic MD subtypes. This topic is underestimated and under investigated and, to our knowledge, combined mutations of *CAPN3* and *DMD* genes were not reported before. In the cases of patient 9, 23 and 24 the suspicion of an overlapping syndrome was based on phenotypic characteristics of the patient that led the clinician to perform further genetic investigations. Indeed, dysferlin and calpain proteins are both essential for the cytoskeleton and physiology of the muscle fiber; hence it is expected that heterozygous mutations, even if not able to cause a clinical phenotype alone, might increase the overall vulnerability to damage of the muscle fiber when associated [37,38]. In the case of association with *DMD*, there are relevant prognostic consequences as cardiac involvement is prominent and frequent in Becker dystrophy [39,40]. Another question is whether alterations of dystrophin may influence the function of calpain. Dystrophin complex plays a fundamental role in the regulations of many enzymatic activities, for example, by maintaining sub-plasma membrane calcium homeostasis. Calcium alteration observed in dystrophin-deficient skeletal muscle cells may interfere with the normal functioning of calpain [37,38]. Further studies are needed to explore these rare association and their consequences on the phenotype.

The use of next-generation sequencing (NGS) panels was essential in achieving a relatively high diagnostic yield in our cohort. NGS allowed simultaneous screening of multiple LGMD genes, increasing the chance of identifying the underlying mutation in these genetically heterogeneous disorders. However, several patients still carried variants of uncertain significance (VUS), which is similar to other NGS-based studies [7,25,26]. This reflects the ongoing challenge of interpreting novel or rare variants, especially in large genes like *TTN* or in genes with complex alternative splicing patterns [35]. Future approaches such as long-read sequencing and RNA transcriptome analysis are likely to play a key role in resolving these uncertain cases, by detecting deep intronic or structural variants that conventional NGS may miss [41]. Establishing an accurate genetic diagnosis is becoming increasingly important, not only for prognosis and genetic counseling, but also because new targeted therapies—such as systemic gene replacement and RNA-based approaches—are being developed and investigated for specific LGMD subtypes. The small size of the cohort and the retrospective nature represents a limitation of the present study. However, it also provides valuable insight into the real-life distribution of LGMD subtypes in a population with unique genetic characteristics and also contributes to enriching the understanding of LGMDs by defining genotype-specific clinical profiles in a Western Sicilian cohort. Future multicenter and longitudinal studies, combined with advanced molecular techniques, are essential to refine genotype–phenotype correlations and support precision medicine initiatives in LGMDs.

## Figures and Tables

**Figure 1 genes-16-00987-f001:**
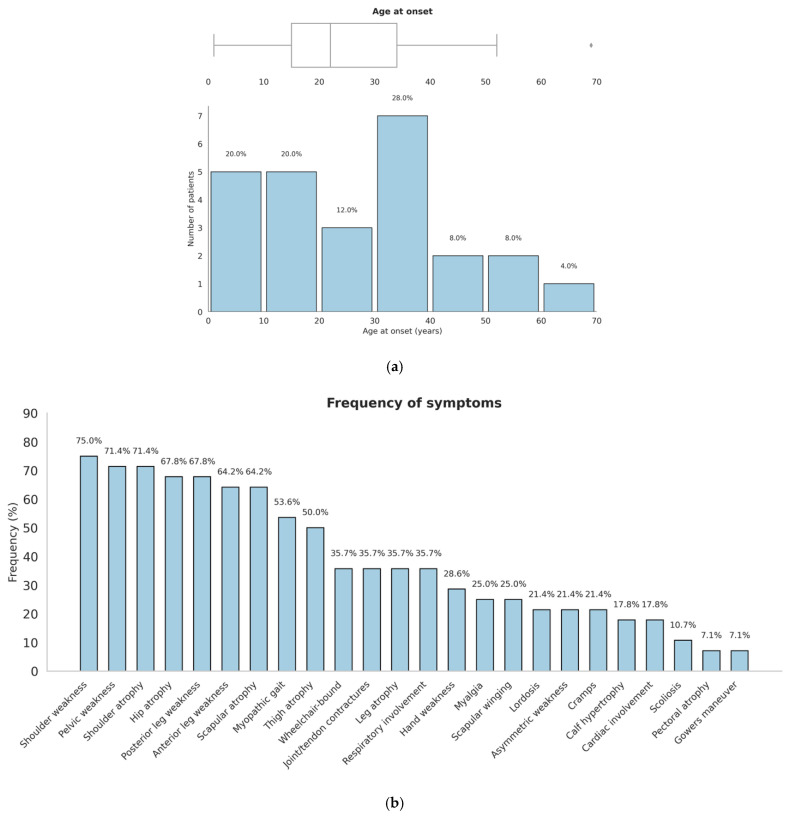
Age at onset and frequency of clinical symptoms in our cohort. (**a**) Distribution of age at disease onset illustrated by a histogram combined with a boxplot. (**b**) Frequency of major clinical symptoms observed in our cohort.

**Figure 2 genes-16-00987-f002:**
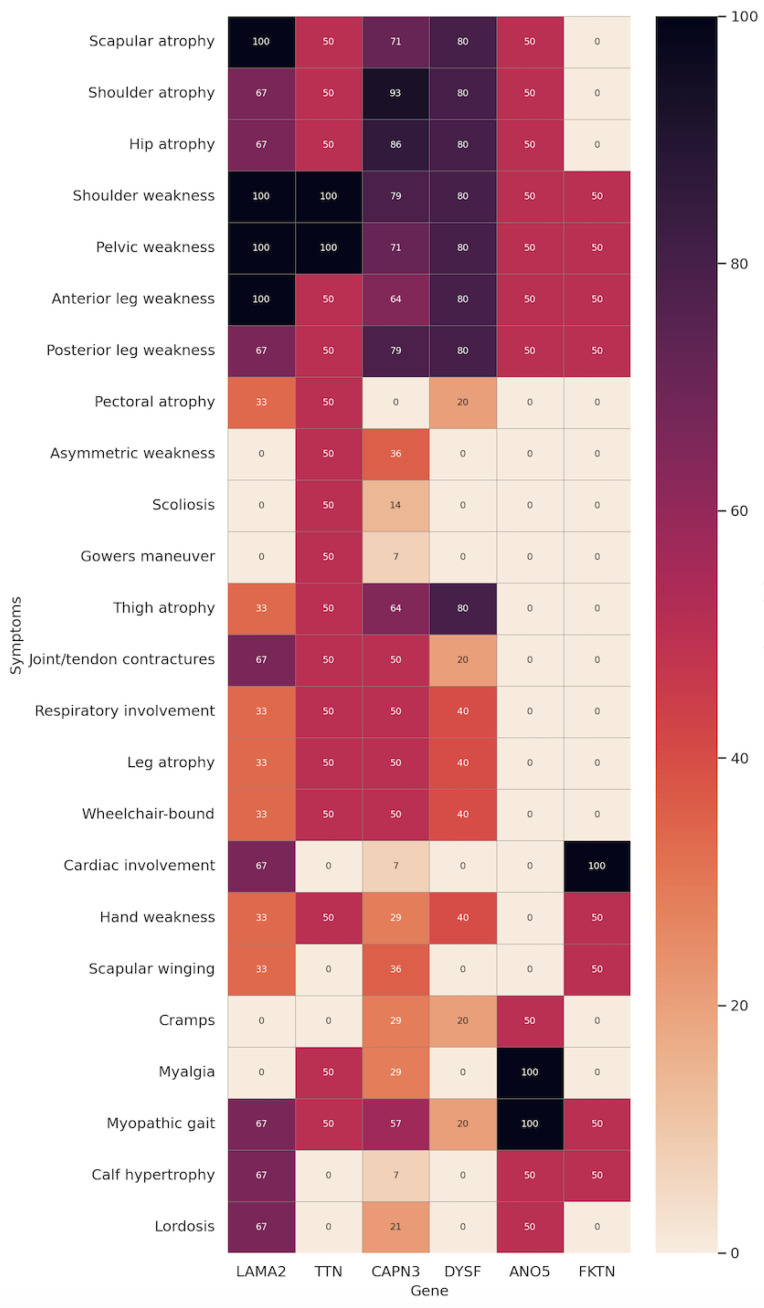
Heat map showing the frequency of individual clinical features across genetic subgroups. This heat map illustrates the distribution of specific clinical manifestations among patients grouped by their causative gene mutation (*LAMA2*, *TTN*, *CAPN3*, *DYSF*, *ANO5*, *FKTN*). Each cell represents the percentage of patients with the indicated symptom within each genetic subgroup, with darker shades corresponding to higher frequencies.

**Figure 3 genes-16-00987-f003:**
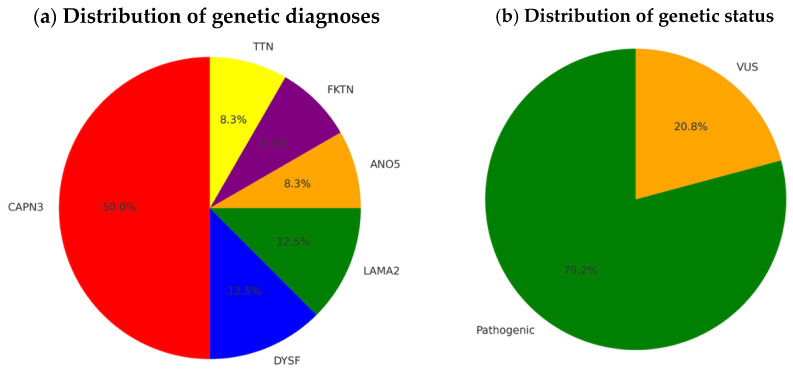
Distribution of genetic diagnoses and variant status in our cohort. The pie chart (**a**) illustrates the distribution of causative genes identified in our patient cohort, with CAPN3 accounting for the largest proportion, followed by *DYSF*, *LAMA2*, *ANO5*, *FKTN*, and *TTN*. The chart (**b**) shows the proportion of pathogenic variants versus variants of uncertain significance (VUS).

**Table 1 genes-16-00987-t001:** Main demographic and clinical characteristics of the LGMD cohort. Continuous variables are expressed as mean with standard deviation if distribution is normal, as median and interquartile range if distribution is not normal and categorical variables as numeric count and percentages. CK, creatin phosphokinase; F, females; IQR, interquartile ranges; M, males; SD, standard deviation.

Clinical Variable	Total (N = 28)
Sex, M/F	16/12
Age, median (IQR)	56.6 (47.2–60.5)
Age at onset in years, mean (SD)	25.6 (17.7)
Disease duration in years, mean (SD)	25.1 (18.1)
Level of CK (U/L), median (IQR)	2000 (521.5–2649)
Muscle biopsy with signs of myopathy, n (%)	11 (39.2%)
Myopathic EMG, n (%)	13 (46.4%)
Muscle RMN, n (%)	1 (3.6%)
Clinical manifestation	
Cramps, n (%)	6 (21.4%)
Myalgia, n (%)	7 (25%)
Asymmetric weakness, n (%)	6 (21.4%)
Shoulder weakness, n (%)	21 (75%)
Pelvic weakness, n (%)	20 (71.4%)
Anterior leg weakness, n (%)	18 (64.2%)
Posterior leg weakness, n (%)	19 (67.8%)
Hand weakness, n (%)	8 (28.6%)
Scapular atrophy, n (%)	18 (64.2%)
Shoulder atrophy, n (%)	20 (71.4%)
Scapular winging, n (%)	7 (25%)
Pectoral atrophy, n (%)	2 (7.1%)
Hip atrophy, n (%)	19 (67.8%)
Thigh atrophy, n (%)	14 (50%)
Leg atrophy, n (%)	10 (35.7%)
Calf hypertrophy, n (%)	5 (17.8%)
Lordosis, n (%)	6 (21.4%)
Scoliosis, n (%)	3 (10.7%)
Joint/tendon contractures, n (%)	10 (35.7%)
Gowers maneuver, n (%)	2 (7.1%)
Myopathic gait, n (%)	15 (53.6%)
Wheelchair dependency, n (%)	10 (35.7%)
Cardiac involvement, n (%)	5 (17.8%)
Respiratory involvement, n (%)	10 (35.7%)

**Table 2 genes-16-00987-t002:** Demographic, clinical and laboratory characteristics stratified by genotype. This table summarizes the main demographic, clinical, and laboratory features of patients in our cohort, grouped according to the causative gene identified (*CAPN3*, *DYSF*, *LAMA2*, *ANO5*, *FKTN*, *TTN*). Continuous variables are expressed as mean with standard deviation if distribution is normal, as median and interquartile range if distribution is not normal and categorical variables as numeric count and percentages. CK, creatin phosphokinase; F, females; M, males; SD, standard deviation.

Clinical Variable	*CAPN3* (n = 14)	*DYSF* (n = 5)	*LAMA2* (n = 3)	*ANO5* (n = 2)	*FKTN* (n = 2)	*TTN* (n = 2)
Sex, M/F	9/5	2/3	1/2	1/1	2/0	1/1
Age, mean (SD)	47.5 (17.4)	55.6 (11.1)	63.3 (15.6)	55 (1.4)	45.5 (14.8)	46 (35.3)
Age at onset in years, mean (SD)	20.5 (13.5)	27.5 (5.9)	19 (23.8)	48.5 (2.1)	32.5 (3.5)	35 (48.1)
Disease duration in years, mean (SD)	26.9 (17.0)	28.2 (3.5)	44.3 (27.7)	6.5 (0.7)	13 (11.3)	11 (12.7)
Level of CK (U/L), mean (SD)	1907.8 (2076.5)	4000 (1414.2)	382.6 (352.1)	2649 (496.4)	2168	751
Muscle biopsy with signs of myopathy, n (%)	6 (42.8%)	1 (20%)	2 (66.6%)	1 (50%)	0	1 (50%)
Myopathic EMG, n (%)	5 (35.7%)	2 (40%)	3 (100%)	2 (100%)	0	1 (50%)
Muscle RMN, n (%)	1 (7.1%)	0	0	0	0	0

**Table 3 genes-16-00987-t003:** Summary of pathogenic variants and VUS identified in the cohort. This table summarizes the molecular results from 24 patients with suspected limb–girdle muscular dystrophy (LGMD) included in our study. For each patient, the identified gene, DNA and protein variants, zygosity, and variant classification (pathogenic or VUS) are reported. VUS, variant of uncertain significance; HOM, homozygosity; HET, heterozygosity; HET.COMP, compound heterozygosity.

Patient	Status	Gene	DNA Variant	Protein	Zygosity
1	VUS	*CAPN3*	c.10G>A	p.Val4Ile	HET
2	VUS	*CAPN3*	c.259C>G	p.Leu87Val	HET.COMP
			c.922G>A	p.Gly308Ser	
3	Pathogenic	*CAPN3*	c.383A>T	p.Asp128Val	HET.COMP
			c.551C>T	p.Thr184Met	
4	Pathogenic	*CAPN3*	c.550del	p.Thr184Argfs*36	HOM
5	Pathogenic	*CAPN3*	c.550del	p.Thr184Argfs*36	HOM
6	Pathogenic	*CAPN3*	c.883_886delinsCTT	p.Asp295Leufs*57	HET.COMP
			c.1792_1795del	p.Lys598Profs*63	
7	Pathogenic	*CAPN3*	c.883_886delinsCTT	p.Asp295Leufs*57	HET.COMP
			c.1792_1795del	p.Lys598Profs*63	
8	Pathogenic	*CAPN3*	c.1303G>A	p.Glu435Lys	HET.COMP
			c.1863_1864del	p.Glu622Glyfs*9	
9	Pathogenic	*CAPN3*	c.1466G>A	p.Arg489Gln	HET
10	VUS	*CAPN3*	c.2257G>A	p.Asp753Asn	HET
11	VUS	*CAPN3*	c.2257G>A	p.Asp753Asn	HET
12	Pathogenic	*DYSF*	c.4194C>A	p.Cys1398*	HOM
13	Pathogenic	*DYSF*	c.4194C>A	p.Cys1398*	HOM
14	Pathogenic	*LAMA2*	c.850G>A	p.Gly284Arg	HOM
15	Pathogenic	*LAMA2*	c.850G>A	p.Gly284Arg	HOM
16	Pathogenic	*LAMA2*	c.1793_1795del	pVal598del	HOM
17	Pathogenic	*ANO5*	c.1664G>T	p.Ser555Leu	HOM
18	Pathogenic	*ANO5*	c.2498T>A	p.Met833Lys	HOM
19	Pathogenic	*FKTN*	c.1304A>G	p.Asp435Gly	HOM
20	Pathogenic	*FKTN*	c.1304A>G	p.Asp435Gly	HET.COMP
			c.1325A>G	p.Asn442Ser	
21	VUS	*TTN*	c.4749_4754del	p.Asn1584_Pro1585del	HET.COMP
			c.49815G>T	p.Lys16605Asn	
22	Pathogenic	*TTN*	c.49948+1G>A		HET.COMP
			c.64397-2del		
23	Pathogenic	*CAPN3*	c.883_886delinsCTT	p.Asp295Leufs*57	COMBINED
			c.1792_1795del	p.Lys598Profs*63	
		*DYSF*	c1385G>A	p.Arg462His	
24	Pathogenic	*DYSF*	c.4194C>A	p.Cys1398*	COMBINED
			c.2051G>A	p.Arg684Gln	
		*CAPN3*	c.590G>A	p.Arg197His	

## Data Availability

The data presented in this study are available on request from the corresponding author due to privacy restrictions.

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
