# Peer review of "Genetic and Clinical Spectrum of Limb–Girdle Muscular Dystrophies in Western Sicily"

_genes, 2025, doi:10.3390/genes16080987_

Round 1

Reviewer 1 Report

Comments and Suggestions for Authors

This is a well-written manuscript supported by data obtained from NGS results. However, I do have some minor comments. Please find below:

  1. Please proofread the manuscript. There are some minor grammatical errors.
  2. Please provide a brief description of the genes involved in LGMD subtypes and their roles in muscle homeostasis. This will help the readers understand the affected genes in more detail.
  3. Line 105: Since all dystrophies are myopathies, when you say myopathies – are you referring to non-dystrophies? Can you please make that clear. This will also be helpful in explaining the serum CK levels of patients who have dystrophies vs non-dystrophies.
  4. Line 219-224: Please change the font size to make it consistent with the rest of the text, since this is not a figure caption.
  5. Line 286: Please add references.

Author Response

Dear Editors and Reviewers

Thanks for your comments and the opportunity to revise our manuscript. Here, we provide a point-to-point answer to all reviewers’ comments. We would like to submit our revised version of the manuscript for possible publication in your journal.

Reviewer 1

This is a well-written manuscript supported by data obtained from NGS results. However, I do have some minor comments. Please find below:

A: Thank you for this careful revision and for your precious and constructive comments. We are grateful for this opportunity to revise and improve our article.

  • Please proofread the manuscript. There are some minor grammatical errors.
    • A: Thank you for this suggestion, we have carefully proofread the entire manuscript and corrected several minor grammatical and typographical issues to improve clarity and language accuracy.
  • Please provide a brief description of the genes involved in LGMD subtypes and their roles in muscle homeostasis. This will help the readers understand the affected genes in more detail.
    • A: Thank you for this important consideration. We have added a brief paragraph in the Introduction section to describe the biological role of the most frequently involved genes (CAPN3, DYSF, LAMA2, ANO5, FKTN, TTN) in maintaining muscle structure and function, and their relevance in muscle membrane repair, calcium homeostasis, sarcolemmal stability. (Lines 48-58, references 8-13).
  1. Chen L, Tang F, Gao H, Zhang X, Li X, Xiao D. CAPN3: A muscle-specific calpain with an important role in the pathogenesis of diseases (Review). Int J Mol Med. 2021;48(5).
  2. Bansal D, Miyake K, Vogel SS, Groh S, Chen CC, Williamson R, et al. Defective membrane repair in dysfer-lin-deficient muscular dystrophy. Nature. 2003;423(6936).
  3. Zhang X, Vuolteenaho R, Tryggvason K. Structure of the human laminin α2-chain gene (LAMA2), which is affected in congenital muscular dystrophy. Journal of Biological Chemistry. 1996;271(44).
  4. Bolduc V, Marlow G, Boycott KM, Saleki K, Inoue H, Kroon J, et al. Recessive Mutations in the Putative Cal-cium-Activated Chloride Channel Anoctamin 5 Cause Proximal LGMD2L and Distal MMD3 Muscular Dys-trophies. Am J Hum Genet. 2010;86(2).
  5. Brockington M, Yuva Y, Prandini P, Brown SC, Torelli S, Benson MA, et al. Mutations in the fukutin-related protein gene (FKRP) identify limb girdle muscular dystrophy 21 as a milder allelic variant of congenital muscular dystrophy MDC1C. Hum Mol Genet. 2001;10(25).
  6. Savarese M, Sarparanta J, Vihola A, Udd B, Hackman P. Increasing Role of Titin Mutations in Neuromuscular Disorders. Vol. 3, Journal of Neuromuscular Diseases. 2016.
  • Line 105: Since all dystrophies are myopathies, when you say myopathies – are you referring to non-dystrophies? Can you please make that clear. This will also be helpful in explaining the serum CK levels of patients who have dystrophies vs non-dystrophies.
    • A: Thank you for this relevant consideration. We agree with the Reviewer’s point and we clarified the terminology in the Methods section. We also replaced the term myopathies with dystrophies in the abstract.
  • Line 219-224: Please change the font size to make it consistent with the rest of the text, since this is not a figure caption.
    • A: Thank you for this suggestion, the font size of the mentioned section has been adjusted to ensure consistency throughout the manuscript.
  • Line 286: Please add references.
    • A: We have added the requested references to support the statement (References: 28-30)
  1. Harris E, Bladen CL, Mayhew A, James M, Bettinson K, Moore U, et al. The clinical outcome study for dysferlinopathy an international multicenter study. Neurol Genet. 2016;2(4).
  2. Nguyen K, Bassez G, Bernard R, Krahn M, Labelle V, Figarella-Branger D, et al. Dysferlin mutations in LGMD2B, Miyoshi myopathy, and atypical dysferlinopathies. Hum Mutat. 2005;26(2).
  3. Ivanova A, Smirnikhina S, Lavrov A. Dysferlinopathies: Clinical and genetic variability. Vol. 102, Clinical Genetics. 2022.

Hoping in positive feedback we look forward to hearing from you soon.

Kind regards,

Vincenzo Di Stefano

Reviewer 2 Report

Comments and Suggestions for Authors

very interesting toipc, but your samplemsize is very small, in adition, tests for differecens are not valid since you dint randomized your patients, fro retrospective single group studies, a multivariate robust for non-normality regression model, based on previous seleccion of varibales based on studies is better

Author Response

Dear Editors and Reviewers

Thanks for your comments and the opportunity to revise our manuscript. Here, we provide a point-to-point answer to all reviewers’ comments. We would like to submit our revised version of the manuscript for possible publication in your journal.

Reviewer 2

very interesting toipc, but your samplemsize is very small, in adition, tests for differecens are not valid since you dint randomized your patients, fro retrospective single group studies, a multivariate robust for non-normality regression model, based on previous seleccion of varibales based on studies is better

  • A: Thank you for this important methodological observation. We are aware that the small sample size and retrospective nature are the main limitations of the study and they surely limit the statistical power of inferential analyses. We have clarified this limitation in the Discussion section of the revised manuscript. More in detail, the primary objective of our study was to better characterize genotype-phenotype correlations in a genetically distinct and historically underrepresented population from Western Sicily. For exploratory purposes, we applied standard statistical comparisons (ANOVA, Mann-Whitney U, and chi-square tests) after verifying distributional assumptions. Despite the mentioned limitations, our study presents several notable strengths. It represents one of the first detailed investigations of limb-girdle muscular dystrophies in a Sicilian cohort, providing novel data from a geographic region with unique genetic background. Indeed, there are scarce reports on genetic data from Sicilians patients and we believe that they are relevant for the scientific community. From a historical point of view, Sicily has been for centuries the center of the Mediterranean area with influences from Eastern, Western, Northern and Southern Countries. Hence, the characterization of such particular geographic area might be of high value for clinical research. The application of a targeted NGS panel led to a high diagnostic yield, allowing us to uncover relevant genotype-specific patterns. Furthermore, we report rare cases of potential digenic inheritance involving CAPN3, DYSF, and DMD genes, expanding the existing knowledge on complex inheritance patterns in LGMDs. We believe that these contributions might enhance the understanding of LGMDs and offer a solid basis for future multicenter and longitudinal studies. 

Hoping in positive feedback we look forward to hearing from you soon.

Kind regards,

Vincenzo Di Stefano

Round 2

Reviewer 2 Report

Comments and Suggestions for Authors

i agree with you about the interest of this research, and your explanations are right but perhaps consider to use multiple regression models with robust version (linear, logistic or multivariate dependin on your dependent variable) to asses wiich factor can provocate differences amnog fenotype group

Author Response

i agree with you about the interest of this research, and your explanations are right but perhaps consider to use multiple regression models with robust version (linear, logistic or multivariate dependin on your dependent variable) to asses wiich factor can provocate differences amnog fenotype group

A: We thank the reviewer for this comment and consideration; we performed a univariate and multivariate logistic binary regression comparing genotypes depending on several covariates, but all the analyses came back statistically inconclusive.